# Insect population dynamics under *Wolbachia*-induced cytoplasmic incompatibility: Puzzle more than buzz in *Drosophila suzukii*

**Alexandra Auguste**[1]*, **Nicolas Ris**[1], **Zainab Belgaidi**[2], **Laurent Kremmer**[1], **Laurence Mouton**[2], **Xavier Fauvergue**[1]

1 ISA, INRAE, Université Côte d'Azur, Sophia Antipolis, France, 2 UMR 5558, Laboratoire de Biométrie et Biologie Evolutive, CNRS, VetAgro Sup, Université de Lyon, Université Claude Bernard Lyon 1, Villeurbanne, France

* alexandra.auguste@inrae.fr

**Data Availability Statement:** All our files are available from the Zenodo database (https://doi.org/10.5281/zenodo.8207425).

## Abstract

In theory, the introduction of individuals infected with an incompatible strain of *Wolbachia pipientis* into a recipient host population should result in the symbiont invasion and reproductive failures caused by cytoplasmic incompatibility (CI). Modelling studies combining *Wolbachia* invasion and host population dynamics show that these two processes could interact to cause a transient population decline and, in some conditions, extinction. However, these effects could be sensitive to density dependence, with the Allee effect increasing the probability of extinction, and competition reducing the demographic impact of CI. We tested these predictions with laboratory experiments in the fruit fly *Drosophila suzukii* and the transinfected *Wolbachia* strain *w*Tei. Surprisingly, the introduction of *w*Tei into *D. suzukii* populations at carrying capacity did not result in the expected *w*Tei invasion and transient population decline. In parallel, we found no Allee effect but strong negative density dependence. From these results, we propose that competition interacts in an antagonistic way with *Wolbachia*-induced cytoplasmic incompatibility on insect population dynamics. If future models and data support this hypothesis, pest management strategies using *Wolbachia*-induced CI should target populations with negligible competition but a potential Allee effect, for instance at the beginning of the reproductive season.

## Introduction

*Wolbachia pipientis* is an endosymbiotic bacterium found in a large number of nematode and arthropod species, including insects [1,2]. The overall record of interactions between *Wolbachia* and their hosts is highly variable, ranging from parasitism to mutualism. The ability of *Wolbachia* to manipulate the reproduction of their hosts at the advantage of its own transmission is the most widespread facet of these interactions [3]. Cytoplasmic incompatibility (CI) is one such manipulation that occurs, in its simplest form, when infected males cross with uninfected females. *Wolbachia*-induced sperm modification in males cannot be restored by females. In diploid species, these crosses result in embryo death while others produce viable offspring [4]. Hence, infected females have more offspring on average than uninfected females

**Funding:** This study was financially supported by the French National Research Agency (ANR-19-CE32-0010). There was no additional external funding received for this study. The funders had no role in study design, data collection and analysis, decision to publish, or preparation of the manuscript.

**Competing interests:** The authors have declared that no competing interests exist.

because they can produce offspring with all types of males, be they infected or not. Since *Wolbachia* is transmitted maternally and vertically, these females are more likely to transmit the incompatible strain, thus promoting its spread [5].

The high invasion success of *Wolbachia* combined with its ability to reduce the transmission of viruses or other pathogens [6,7] has aroused the interest of epidemiologists. Disease-vector control strategies have been developed based on the introduction of individuals artificially infected with a virus-blocking incompatible strain of *Wolbachia* into a target population. The reproductive advantage of infected females favors the invasion of the virus-blocking strain, which increases in frequency until it becomes the majority or fixed strain. In the event of a resident strain not inducing CI is present in the population, it would be replaced by the introduced incompatible strain, which results in a host population infected with a *Wolbachia* strain with traits of interest for pest control. For instance, such a replacement strategy has been shown to be effective in suppressing dengue transmission by the mosquito *Aedes aegypti* [8].

In order to optimize the success of such *Wolbachia* replacement programs, it is necessary to identify the factors that influence *Wolbachia* invasion dynamics. Historically, most theoretical models of CI dynamics are analogous to population genetic models and have focused on the evolution of the frequencies of an incompatible strain within the host population, neglecting the specificities of host population dynamics [9–13]. Only a limited number of host demographic parameters directly affecting changes in frequencies were considered and assumed to be constant [8,13,14]. These models predict the existence of a critical infection frequency, referred to as the invasion threshold, above which an incompatible strain will deterministically invade the host population. The threshold depends on three parameters: the intensity of infection-induced CI, the success of maternal transmission and the costs resulting from infection, such as reduced fertility. More recently, Hancock *et al.* [15–17] added density dependence on host demographic parameters in order to study *Wolbachia* invasion dynamics with more realistic assumptions. They concluded from different models that the intensity of competition, by varying the reproductive success of individuals, significantly reduces the ability of *Wolbachia* to invade a population [16,17].

The ability of *Wolbachia* to invade may serve another pest management strategy, alternative to the drive of interesting characters. In a landmark publication, Dobson *et al.* [18] have proposed a novel CI-based insect pest suppression strategy based on *Wolbachia* invasion. They extended classical CI models by assuming realistic host population dynamics with density dependent survival of immature stages and parameter values taken from the insect literature. They simulated the introduction of an incompatible *Wolbachia* strain into a host population that had reached carrying capacity. While the incompatible strain spreads through the population, the frequency of incompatible crosses increased, resulting in a transient reduction in host population size. Population size declined to a minimum when the proportions of infected and uninfected individuals were similar and then increased again when the incompatible strain became the majority. One of their predictions is that the minimum population size reached during invasion is influenced not only by *Wolbachia*'s own invasion parameters, but also by demographic parameters intrinsic to host populations. For instance, under conditions of high host reproductive rate and intraspecific competition, the demographic consequences of CI may go unnoticed.

The transient population decline caused by *Wolbachia*-induced cytoplasmic incompatibility, as predicted by Dobson *et al.* [18], is a form of demographic destabilization that, in conjunction with other processes, could cascade towards population extinction. As other process, the Allee effect is an excellent candidate. It is characterized by population growth rate decreasing with the decrease of the population size, which triggers an extinction vortex if the growth rate becomes negative below a critical number of individuals (the Allee threshold) or if the growth rate remains positive but combines with stochasticity. The Allee effect has thus been

envisioned as a catalyst for extinction upon which novel tactics could be developed to eradicate pest populations [19–23]. Building on these ideas, Blackwood *et al.* [24] assumed an Allee effect to investigate the conditions under which CI expression would serve pest eradication. To this end, they simulated *Wolbachia* introductions into populations affected by a more or less intense Allee effect. When the Allee effect was strong and the population reproduction rate low, the model predicted that the minimum population size reached during *Wolbachia* invasion would fall below the Allee threshold, leading to population extinction. On the other hand, at high reproduction rates, the transient reduction induced by CI was not large enough to bring the population size below the Allee threshold.

The theoretical models of Dobson *et al.* [18] and Blackwood *et al.* [24] are of major interest for insect pest management but, to our knowledge, they have not been tested yet. We are unaware of any empirical evidence of either transient population declines caused by the invasion of an incompatible *Wolbachia* strain and subsequent increase in CI or population extinctions if CI combines with an Allee effect. We therefore initiated a research program using these models as a general theoretical framework and *Drosophila suzukii* as a model organism.

There are several reasons for choosing *Drosophila suzukii* as a model organism. First, *D. suzukii* is a major invasive pest of berry and stone fruit crops [25]. Native from Southeast Asia, *D. suzukii* was first reported in the USA and Europe in the late 2000s and continues to spread worldwide (*e.g.* Africa; [26–28]). Unlike other *Drosophila* species, the serrated ovipositor of *D. suzukii* allows females to lay eggs under the skin of ripening or mature fruits, causing significant damage to cultivated fruits and consequently, huge economic losses [25]. This calls for research to develop environmentally friendly methods to control this major agricultural pest. Second, *D. suzukii* is naturally infected with *Wolbachia*. Contrary to the common observation of "most-or-few" infection frequencies, the prevalence of the resident *w*Suz strain in European populations is highly variable, averaging 46% [29]. This infection polymorphism could be a consequence to the fact that *w*Suz induced no or weak CI [29,30] as well as the variable benefits and costs of *w*Suz protection from viruses [31]. Third, we capitalized on a strong body of experimental work based on the transinfection of *D. suzukii* with various *Wolbachia* strains, including *w*Ha and *w*Tei, which cause cytoplasmic incompatibility [32–34]. Finally, the expression of negative density dependence on individual performance has already been observed in *D. suzukii* [35]. Under experimental conditions with high larval densities, some individuals were observed to leave the breeding site, which was interpreted as a response to intraspecific competition for resources [36]. While the Allee effect has not yet been demonstrated in *D. suzukii*, it has been documented in other closely related *Drosophila* species [37].

The aim of this study is to estimate the strength of positive and negative density dependence in *D. suzukii* and to test the effect of *Wolbachia*-induced CI on the dynamics of experimental populations. In this article, we report the absence of an Allee effect under our laboratory conditions. In contrast, we found strong negative density dependence resulting in logistic population growth. Finally, we found no effect of introducing an incompatible *Wolbachia* strain on *D. suzukii* population dynamics. We discuss these results in the light of the models developed by Dobson *et al.* [18] and Blackwood *et al.* [24] and propose that density dependent competition may have much stronger influences than expected on *Wolbachia* invasion and the demographic consequences of cytoplasmic incompatibilities.

## Materials and methods

### Insect rearing

Two lines of *D. suzukii* have been maintained in the laboratory since 2018, *i.e.*, for around 24 generations before experiments. These lines were infected with either *w*Suz, the strain

naturally present in *D. suzukii* [29,30,38] or the transinfected *Wolbachia* strain *w*Tei, the natural symbiont of *D. teissieri*. The latter was selected because of the incompatibility induced by transinfected *D. suzukii* males when crossed with *w*Suz-infected females [32,34]. The two lines were reared on cornmeal-based food medium [39] containing, per 1 L of water: 10 g agar, 80 g corn flour, 80 g brewer's yeast and 60 ml moldex (100 g methyl hydroxybenzoate in 1 L 70% ethanol). Rearing, as well as all experiments, occurred in a built-in climatic chamber at 23–24˚C, 50–60% humidity and a photoperiod of 12 h-12 h.

## Experiment 1. Estimation of population size

In order to estimate accurately the number of flies in population cages, we developed a rapid photo-based census method. Prior to experiments, we assessed the relevance of this method. For this, between 4 and 396 individuals were introduced into 59 cubic experimental cages (21 cm × 21 cm × 21 cm) made of fine-mesh fabric and a transparent plastic front panel. The cages were then photographed using a Nikon N750D camera and a Nikon lens (AF-S Micro Nikkor 60 mm f/2.8G ED). Visible Individuals were counted on a computer using the software ImageJ 1.52 [40] on three predefined areas (S1 Fig): (i) individuals appearing on the front and on the rear panels (area 1), (ii) individuals appearing on the whole surface except the left side and the cage floor (area 2), or (iii) only individuals appearing on the front panel (area 3). Areas 1 and 2 were used for the experiments 2 and 3, respectively. Area 3 was not used in the present study but it will be used in follow-up articles. The reasons for this choice are given in the Results section.

For each area, linear regressions between the numbers of individuals on pictures and the total numbers in the cage were fitted, and the coefficients of determination $R^2$ estimated. Kolmogorov-Smirnoff tests, from the DHARMa package in R, were used to test normality of the residual distribution. These and the following statistical analyses were carried out using R 4.2.2 software [41].

## Experiment 2. Early dynamics of small populations

**Founding populations.**  This experiment aimed at quantifying a demographic Allee effect, *i.e.*, positive density dependence at low population size. To this end, we founded *D. suzukii* populations with different numbers of male-female pairs and estimated subsequent growth rates. Founding pairs were formed with 1 male and 1 female, all virgins and less than 24 hours old. Populations were founded with either 1, 3, 10, 30 or 100 pairs, and each of these initial population size was replicated between 8 and 13 times (13 populations with 1 pair, 13 with 3 pairs, 10 with 10 pairs, 8 with 30 pairs, and 8 with 100 pairs, for a total of 52 populations). Replicated populations were distributed across four blocks spaced one week apart. For each population, the pairs were introduced into a cage as described above. Weekly, flies were provided with a new oviposition patch consisting of 100 ml (75 mm diameter) rearing medium supplemented with a fresh raspberry. As the generation time was around two weeks in our rearing conditions, each patch was exposed three weeks to ascertain that eggs laid in the patch yielded new adults emerging into the cages. Immature drosophila developed by feeding on the rearing medium whereas adult flies were supplied with water and honey three times a week. To estimate population size, each cage was photographed once a week. Populations were followed nine weeks post foundation; generations were completely overlapping, but if they were discrete, nine weeks would represent 4 to 5 generations of *D. suzukii*.

**Fitting an exponential growth model.**  The early dynamics of populations founded with a small number of individuals is expected to depend on initial population size and the strength of an Allee effect. In the absence of an Allee effect, exponential population growth is expected.

With a weak Allee effect, exponential growth is also expected, but the growth rate should be lower for initially small populations (*e.g.*, 1 pair, 3 pairs, etc.) than for larger ones (*e.g.*, 100 pairs). With a strong Allee effect, populations of initial size smaller than the Allee threshold should go extinct whereas populations initially above the Allee threshold should increase exponentially, with a growth rate again depending on population size. We tested these predictions by counting populations that went extinct and estimating the growth rate of those that persisted. The growth rate (*r*) was estimated by fitting a simple exponential growth model such as:

$$N_t = N_0 e^{r\,t} \tag{1}$$

to each population time series, excluding the last week at which the largest populations had stopped increasing. In practice, we fitted weekly counts with a generalized linear model based on a Poisson distribution and a log link function, *i.e.*:

$$\log(N_t) = \log(N_0) + r\,t \tag{2}$$

The initial number of individuals observed just after the populations had been founded was defined as an offset ($\log(N_0)$), that is, a parameter included in the model but not estimated. The number of individuals estimated immediately after release (week 1) was missing for all populations from the first block. It was impossible to add the initial number as an offset to the model. Therefore, the five populations of the first block were discarded due to missing values. Growth curves and estimated growth rates *r* were obtained for each population. A linear regression was then used to analyze the effect of the number of founders on growth rate.

**Testing density dependence.**   A complementary method to test for positive density dependence is to assess the relation between the proportion *p* of populations growing between *t* and *t + 1* and the population size at time *t*. Analyzing *p* rather than individual counts circumvents the high sampling variance at low numbers and the consequent spurious estimations of growth rates [42]. With an Allee effect, we expect *p* to increase with population size at low population size, with *p* = 0.5 being the Allee threshold. Independently of the experimental manipulation of initial population size, we associated all possible pairs of $N_t$ and $N_{t+1}$ for each of the 47 populations at each weekly interval. A population was considered stable or growing for $N_{t+1} \geq N_t$ (further referred to as a growing population given the low number of pairs, 14 of 410 pairs, with $N_{t+1} = N_t$) or decreasing otherwise. We then calculated the proportion *p* of growing populations within 18 classes of population sizes (1–49, 50–99, etc.). This proportion was fitted with a logistic regression to test three alternative hypotheses: (i) a first null model suggests no relation between population size and growth

$$p = \beta_0, \tag{3}$$

(ii) a second standard model suggesting a linear relation

$$p = \beta_0 + \beta_1 N_t, \tag{4}$$

and (iii) a third polynomial model allowing a non-linear relationship

$$p = \beta_0 + \beta_1 N_t + \beta_2 N_t^2 \tag{5}$$

The Eq (5) is well suited to describe bell-shaped curves that characterize populations with both an Allee effect and negative density dependence. The Akaike Information Criterion (AIC) was used to select the best model.

## Experiment 3. *Wolbachia* invasion and consequent population dynamics

**Experimental populations.** The aim of this experiment was to assess changes in *D. suzukii* population dynamics caused by the introduction of individuals infected with an incompatible *Wolbachia* strain. To this end, we founded eight populations with 100 *w*Suz-infected individuals (sex ratio 1:1) aged 3 to 5 days. Each population was studied continuously across three distinct periods, for a total of 23 weeks: a first period of population growth (5 to 8 weeks); a seven-week period of dynamic equilibrium around the carrying capacity, starting upon observation of the first decrease in number; a post-introduction period (8 to 11 weeks) starting upon the introduction of infected flies. Populations were maintained as in the previous experiment, with a new reproduction patch provided weekly and exposed during three weeks, and water and honey to feed adults three times a week. Cages were again photographed every week for estimation of population sizes.

**Introducing *Wolbachia* and characterizing infection status.** Individuals introduced after the seven-week period of dynamic equilibrium were infected by either *w*Suz for the four "control" populations (without cytoplasmic incompatibility) or *w*Tei for the four "CI" populations (with cytoplasmic incompatibility). Theoretically, *Wolbachia* invasion depends on the intensity of CI, transmission efficiency, and infection cost [9–13]. Assuming perfect transmission, an incompatible strain should invade if it is introduced above a threshold equal to the ratio of infection cost (parameter $sf$ in the models referenced above) to incompatibility (parameter $sh$). In our system of *D. suzukii* and *w*Tei, incompatibility is incomplete, with an egg hatch rate of about 33% after incompatible crosses (*i.e.*, $sh$ = 0.67; [34]). In parallel, the analyses of traits such as fecundity, longevity, egg hatch rate and developmental time all suggest an absence of cost of *w*Tei infection on *D. suzukii* [32,34]. This apparent absence of infection costs ($sf$ = 0), even with incomplete incompatibility, suggests a theoretical invasion threshold of 0%. In practice, this would mean just enough individuals to compensate for the stochastic events that are not considered in the theoretical models. A number of individuals close to 10% of the mean number of individuals over equilibrium period should be sufficient to trigger invasion. Before introduction, we verified the fixation of *w*Suz for all populations by genotyping 20 individuals per population (methods below). After introduction, we monitored the evolution of *w*Tei frequencies over time in the four CI populations. For this, we randomly sampled 20 individuals per week and stored them in 99% ethanol at—20˚C until analysis. Individual infection status was assessed *via* HRM (High Resolution Melting), a genotyping method that allows the detection of changes in the melting profiles of double-stranded DNA (S2 Fig, [43]). For each individual, DNA was extracted using Macherey Nagel's NucleoSpin kit. *Wolbachia*-specific primers 81f (5′–TGGTCCAATAAGTGATGAAGAAAC–3′; [44]) and 2R (5′–CAGCAATTTCAGGATTAG–3′; [43]) were used to amplify the *wsp* ("*Wolbachia Surface Protein*") gene. PCRs were performed in a 10 μL volume containing 500 nM primers, 1X Precision Melt Supermix (Biorad), containing dNTPs, iTaqTM DNA polymerase, MgCl2, EvaGreen buffer, stabilizers, and 2 μL of 10-fold diluted DNA [43]. Cycle conditions were 95˚C for 120 s, then 30 s at 95˚C, 30 s at 54˚C and 30 s at 72˚C for 40 cycles, followed by 30 s at 95˚C and 60 s at 60˚C (Biorad CFX96). The temperature ramp rate was 1.6˚C/s.

**Estimating carrying capacity.** The eight populations were studied over enough time to estimate the carrying capacity $K$ in our environmental conditions. For this, we fitted non-linear regressions to the number of adults observed each week, from foundation to the introduction of infected flies (first and second time periods). Three candidate population growth models were fitted: Gompertz, logistic, and Weibull. SSlogis, SSgompertz and SSweibull functions in R were used to obtain the initial parameter values for each candidate model (so-called "selfStart" models [41,45]). All these models share one parameter, the asymptotic population

size defined as the carrying capacity. The AIC and root-mean-square error (RMSE) were then used to select the best of the three models fitted to each population.

**Testing the consequences of CI on population dynamics.** Evolution of the frequency of *w*Tei-infected individuals was analyzed using a generalized linear mixed model based on a negative binomial distribution and a logit link function, with time as a fixed effect and population as a random effect. We excluded the first period of population growth to focus on the influence of introducing incompatible *Wolbachia* on the dynamics of populations that had reached carrying capacity. The main prediction tested here is that the introduction of individuals infected by the incompatible *w*Tei *Wolbachia* strain should lead to a decline in population size, co-occurring with the invasion of *w*Tei and subsequent increase in the proportion of incompatible crosses. This decline should not be observed in control populations with introductions of individuals infected by *w*Suz. We thus fitted the following model:

$$N_{ijt} = \beta_0 + \beta_1\ t + \beta_2\ intro + \beta_3\ t\ intro + \beta_4\ t\ intro\ strain + \varepsilon_{ijt} \tag{6}$$

where $N_{ijt}$ is the number of individuals in populations of introduction status *i* (before *vs* after introduction; variable *intro* in the model) having undergone treatment *j* (introduction of *w*Tei or *w*Suz; variable *strain* in the model) at time *t* relative to the introduction (rescaled so that *t* = 0 upon introduction). This model allowed to test four predictions (S3 Fig).

H1: Control and CI populations were at equilibrium before introduction so that there should be no effect of time during the seven weeks prior to the introduction ($\beta_1 = 0$).

H2: Introductions should result in an immediate but small increase in population size ($\beta_2 \neq 0$).

H3: Introductions may destabilize populations from equilibrium and hence, induce a change in the effect of time on population size ($\beta_3 \neq 0$).

H4: The latter change is expected to be affected by the strain, with *w*Tei, contrary to *w*Suz, inducing a decrease in population size ($\beta_4 \neq 0$).

These four hypotheses were tested with likelihood ratio tests between models with and without the variable or interaction concerned.

## Results

### Estimation of population size (experiment 1)

In order to estimate population size, we counted adult flies on pictures of three different areas of the experimental cages. Whatever the area considered, we found a clear relationship between the number counted on pictures and the total number of flies in the cages (Fig 1). Linear regressions were characterized by high coefficients of determination: $R^2 = 0.99$ for areas 1 and 2 and $R^2 = 0.86$ for the last area. Kolmogorov-Smirnov tests confirmed the normal distribution of errors, and hence, an absence of sampling bias (Fig 1; KS test: $p = 0.83$, 0.85 and 0.34 for areas 1, 2 and 3 respectively). Given the high coefficients of determination obtained for areas 1 and 2 and the small number of populations involved, these two areas were chosen to estimate the population size in experiments 2 and 3 respectively. However, should the number of experimental populations increase, we would recommend the use of area 3, because counting higher numbers of insects in the other two areas would be too time-consuming to characterize many populations on a weekly basis. Therefore, area 3 was not used in the present study, but it will be used in the future.

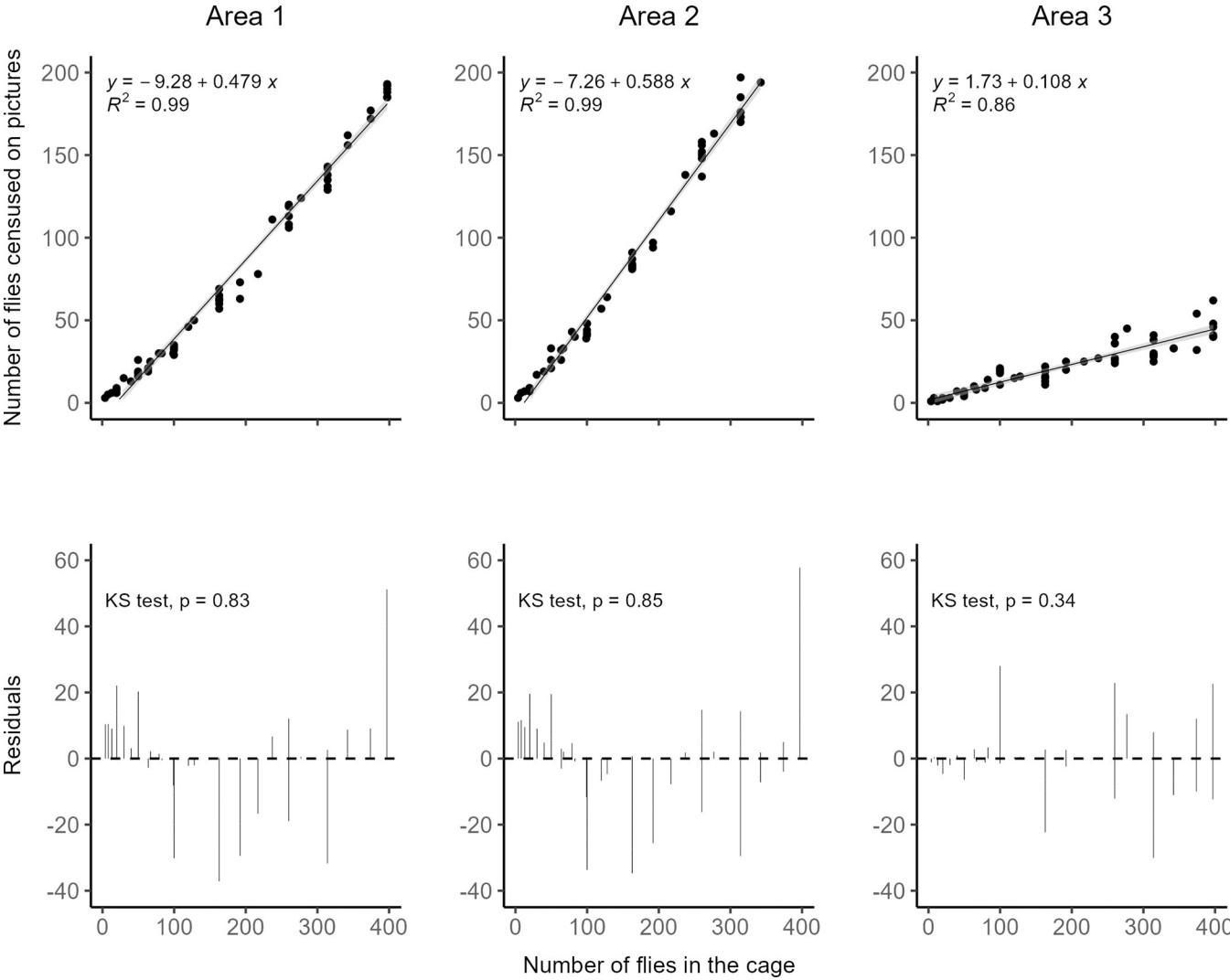

**Fig 1. Estimation of population size.** Linear regressions between the number of adult *D. suzukii* censused on pictures of three different areas of the experimental cages and the total number of individuals in the cage.

### Early dynamics of small populations (experiment 2)

None of the newly founded populations went extinct, even introducing a single pair. Population growth during the first eight weeks was well described by an exponential model, regardless of the number of founding pairs (Fig 2). The estimated growth rate *r* was influenced by initial population size (Anova, $F_{4,42} = 14.6$, $p < 0.001$). Contrary to expectation with an Allee effect, the growth rate *r* was highest at smallest population sizes, from $1.73 \pm 0.03$ for single pairs to $1.4 \pm 0.04$ for 100 pairs.

The detailed analysis of week-to-week variations in population sizes confirmed the absence of Allee effect. Among the three alternative models, the second order polynomial logistic regression best fitted the proportion of increasing populations (AIC = 61.3, *versus* 82.9 for the null model and 63.5 for linear regression). The resulting model (logit $p = 1.814 + 2.014 \; 10^{-3} N_t - 1.056 \; 10^{-5} N_t^2$) suggests that the proportion of populations growing between consecutive weeks (*p*) peaks at a population size of about $N_t = 95$ individuals. This is the population size at

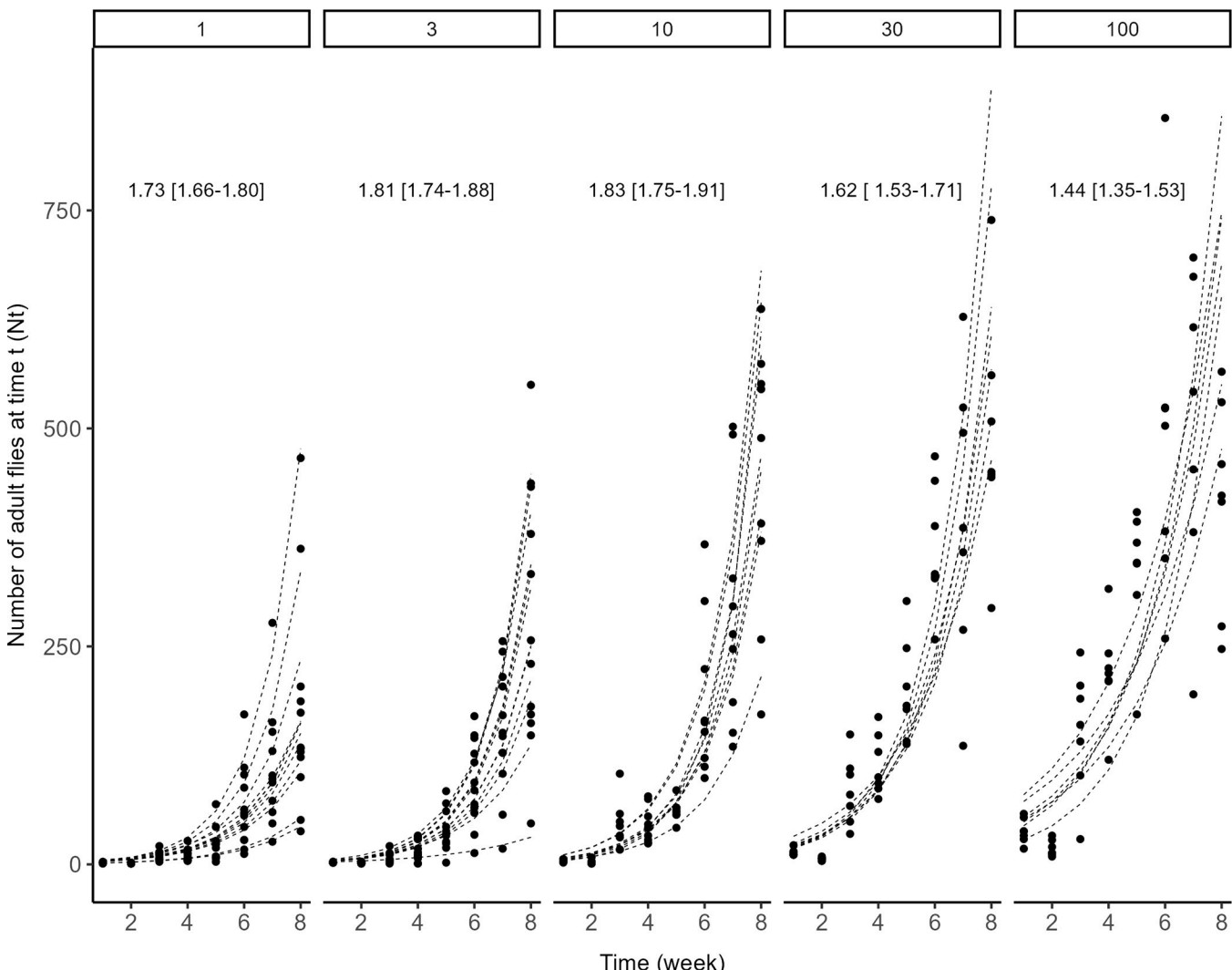

**Fig 2. Early dynamics of small *D. suzukii* populations initiated with 1, 3, 10, 30 or 100 male-female pairs.** For each replicated population, the eight-week time series of observed number of individuals (circles) was fitted with an exponential growth model $N_t = N_0 \, e^{\, r \, t}$ (lines). Estimated marginal means of population growth rate $r$ and their 95% confidence intervals are shown.

which the derivative of the above equation equals zero. However, the predicted value of $p$ for this "optimal" population size ($p = 0.87$) is not different from that predicted for $N_t = 0$, *i.e.* the intercept: $p = 0.86$ [0.77–0.92]. Thus, despite a second-order logistic regression best fitting the data, there is no evidence of an Allee effect. This also appears clearly in Fig 3.

Altogether, these results suggest an absence of Allee effect in our conditions. Rather, the decrease in population growth with increasing population size reflects negative density dependence.

## Reaching carrying capacity (Experiment 3)

The eight experimental populations studied over a longer time period were well described by Gompertz, Logistic or Weibull population growth models (Table 1). Notwithstanding slightly different shapes, these three models are characterized by a plateau, the carrying capacity $K$ (Fig 4). Although the estimated values of $K$ varied between populations, they were of similar order

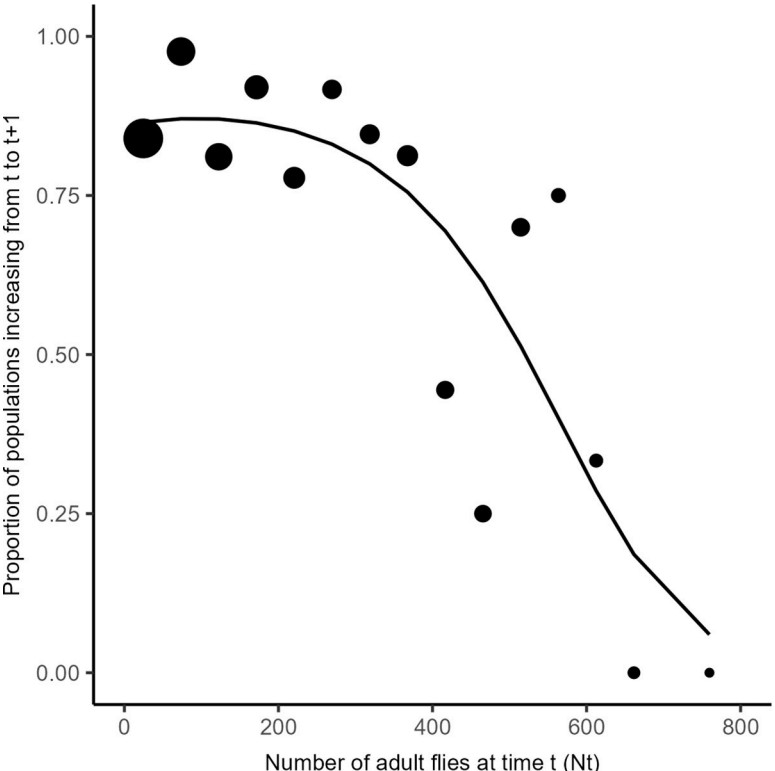

**Fig 3. Proportion of the populations increasing ($t$ to $t$ + 1).** For each range of population size at time $t$ ($N_t$: 0–49, 50–99, etc.) circles represent the proportion $p$ of pairs for which the number of individuals increased ($N_{t+1} \geq N_t$). Circle size represent the number of pairs available to estimate each proportion (largest circle: x pairs; smallest circle: y pairs). The curve represents the second-order polynomial regression that best fitted the data (logit $p$ = 1.814 + 2.014 $10^{-3}$ $N_t$— 1.056 $10^{-5}$ $N_t^2$).

of magnitude, between 455 and 593 (Table 1). This carrying capacity was reached after five to eight weeks.

## Consequences of cytoplasmic incompatibilities at carrying capacity (Experiment 3)

To test the effect of introducing CI into a recipient population of *D. suzukii*, we fitted a generalized linear model to population size, with a negative binomial distribution of errors and a log link function (Fig 5). The variance in numbers within replicate was low (5.3 $10^{-12}$) so we added no random effect. Hypotheses H1 to H4 were tested *via* likelihood ratio tests for comparisons of models with and without parameters under scrutiny.

H1: as expected from the design of the experiment, prior to the introductions of infected flies, the numbers of individuals varied at random around the carrying capacity ($\beta_0$ = 6.21; test for parameter $\beta_1$ = - 0.02; $LR_{1,135}$ = 0.98, $p$ = 0.32).

H2: as expected, introducing infected flies had an immediate, positive influence on population size (test for parameter $\beta_2$ = 0.24; $LR_{1,135}$ = 6.13, $p$ = 0.01).

H3: following the immediate increase in numbers, the introduction of infected individuals had no general influence on the variation over time (test for parameter $\beta_3$ = - 0.009; $LR_{1,135}$ = 0.18, $p$ = 0.67).

**Table 1. Gompertz, logistic and Weibull population growth models fitted to each of the eight experimental population and values of Akaike Information Criterion (AIC), Root-Mean-Square Error (RMSE) and carrying capacity *K*.**

| Population | Model | AIC | RMSE | K estimate |
|---|---|---|---|---|
| 1 | Gompertz | 197.9 | 135.7 | 597.2 (452.6,741.8) |
| | **Logistic** | **197.3** | **133.3** | **593.5 (466.3,720.6)** |
| | Weibull | _ | _ | _ |
| 2 | **Gompertz** | **164.2** | **162.1** | **455.7 (317.1,594.3)** |
| | Logistic | _ | _ | _ |
| | Weibull | _ | _ | _ |
| 3 | Gompertz | _ | _ | _ |
| | Logistic | 154.7 | 109.4 | 572.7 (478.5,666.9) |
| | **Weibull** | **155.7** | **104.9** | **570.7 (475.5,665.9)** |
| 4 | Gompertz | 165.3 | 66.6 | 489.0 (433.7,544.3) |
| | Logistic | _ | _ | _ |
| | **Weibull** | **158.1** | **47.9** | **488.2 (449.6,526.8)** |
| 5 | Gompertz | _ | _ | _ |
| | Logistic | _ | _ | _ |
| | **Weibull** | **169.0** | **109.6** | **527.4 (436.2,618.7)** |
| 6 | Gompertz | 182.6 | 81.5 | 543.3 (464.9,621.7) |
| | **Logistic** | **181.7** | **79.2** | **533.4 (467.4,599.4)** |
| | Weibull | 183.4 | 78.2 | 523.8 (462.0,585.5) |
| 7 | Gompertz | 179.5 | 110.6 | 520.0 (400.2,639.9) |
| | Logistic | 178.6 | 106.9 | 510.4 (415.5,605.3) |
| | **Weibull** | **177.0** | **94.2** | **504.5(427.3,581.7)** |
| 8 | Gompertz | 185.2 | 135.7 | 579.3 (459.6,699.0) |
| | **Logistic** | **184.7** | **133.4** | **577.6 (464.9,690.3)** |
| | Weibull | _ | _ | _ |

*K* (referred as *Asym* in the models) is the parameter common to all three models and that corresponds to the asymptotic value of the population size when time tends to ∞, or the carrying capacity. Values of *K* and 95% confidence intervals were estimated. Models in bold are those that best fit the observed data (lowest AIC and RMSE).

H4: most importantly, this absence of effect of introduction on population dynamics was consistent for the two strains of *Wolbachia*, *i.e.*, the interaction *time × strain* after introduction was not significant (test for parameter $\beta_4$ = - 0.02; $LR_{1,135}$ = 3.11, $p$ = 0.08). A trend in the expected direction can be observed and could be responsible for this marginally significant test, but this trend seems to be partly driven by only few data points (populations 6 and 8; Fig 5).

Given the importance of this latter test and the low *p*-value, we complemented the parametric analysis with a chi-square test to compare the proportion of weekly observations where population sizes were below and above the carrying capacity. Before introduction, 53% (15/28) and 64% (18/28) observations were below *K* for control and CI populations respectively, and these proportions did not differ ($\chi^2$ = 0.30, $p$ = 0.59). After introduction 38% (14/37) observations were below *K* for CI populations, a proportion that, contrary to theoretical predictions, was not higher than that observed for control populations (47%, 18/38; $\chi^2$ = 0.36, $p$ = 0.55).

Altogether, these results suggest that introducing the incompatible *Wolbachia* strain *w*Tei into a population of *D. suzukii* at carrying capacity produced an immediate and limited burst in population size but no consistent decrease in population size over time.

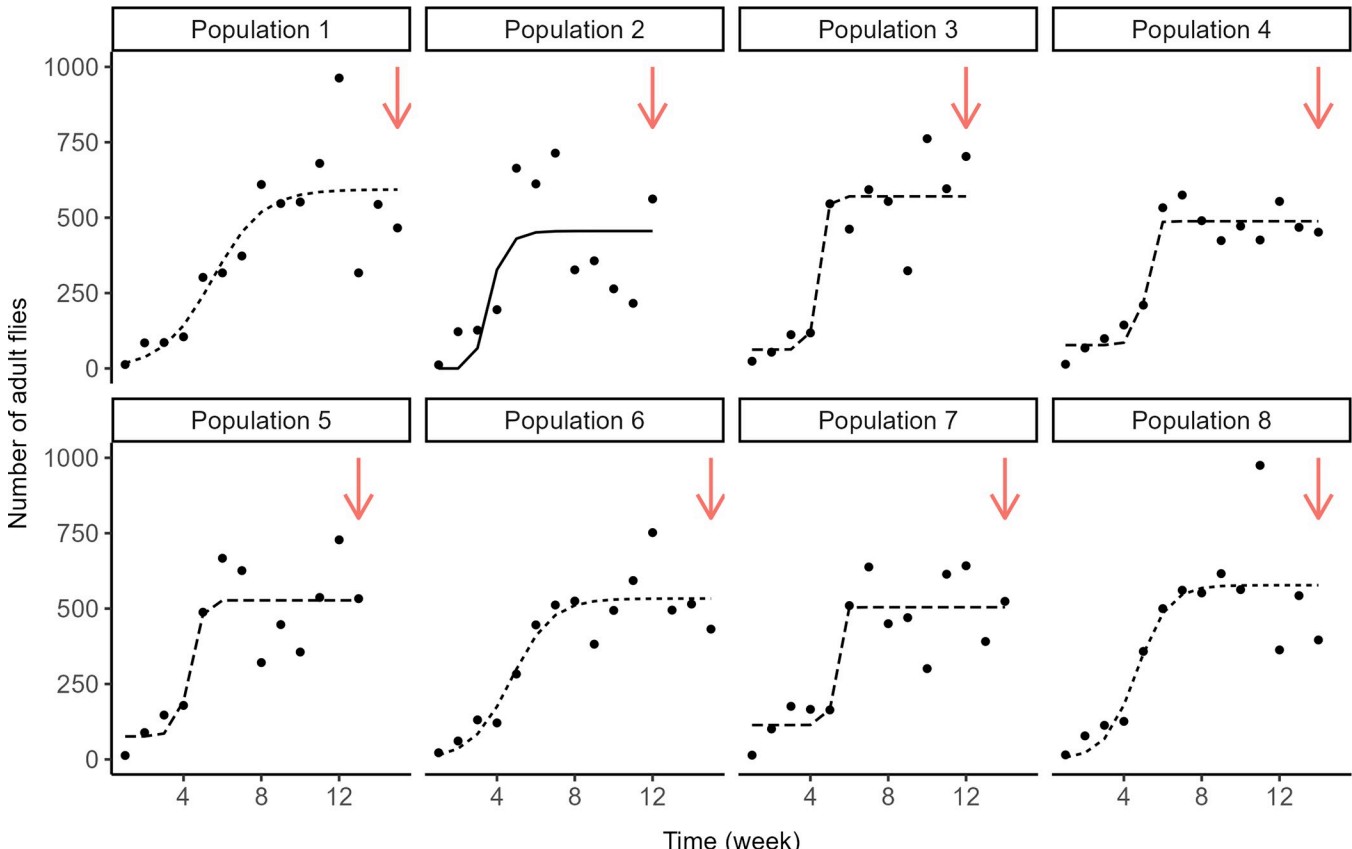

**Fig 4. Reaching the carrying capacity.** Observed and predicted time variations of *D. suzukii* population size for eight replicated populations before the introduction of *Wolbachia*-infected individuals. Line types represent the logistic population growth model that best fitted the data (dashed = Weibull, solid = Gompertz, dotted = logistic). Red arrows indicate the time at which infected flies were introduced, *i.e.*, seven weeks after the first observed decrease in population size.

### Invasion dynamics of the incompatible strain (Experiment 3)

After introducing flies infected by *w*Tei at an initial frequency of 10% of the estimated population size, the prevalence of *w*Tei did not change consistently over time ($\chi^2$ = 3.55, *df* = 1, *p* = 0.06). The prevalence of *w*Tei reached up to 25% in two populations, which probably produced the marginally significant test, but it was below the initial frequency introduced in the other two (Fig 6). For populations with weak evidence for invasion of *w*Tei, weekly variations in the frequency of infected individuals were high, which could reflect random effects such as sampling variance, drift, or the combination of both processes.

### Discussion

The endosymbiotic bacterium *Wolbachia pipientis*, widespread in insects, often causes cytoplasmic incompatibility, a reproductive failure that occurs when males and females do not have the same infection status [3]. This has led to novel and interesting ideas for insect pest management based on the introduction of an incompatible *Wolbachia* strain into a target population, with the assumption that the resulting incompatibilities will affect insect population dynamics. The reasoning is that if the incompatible strain is introduced at an initial frequency above its invasion threshold, it will invade the population, which should then lead to an increase in the proportion of incompatible crosses and, in turn, a decrease in the population

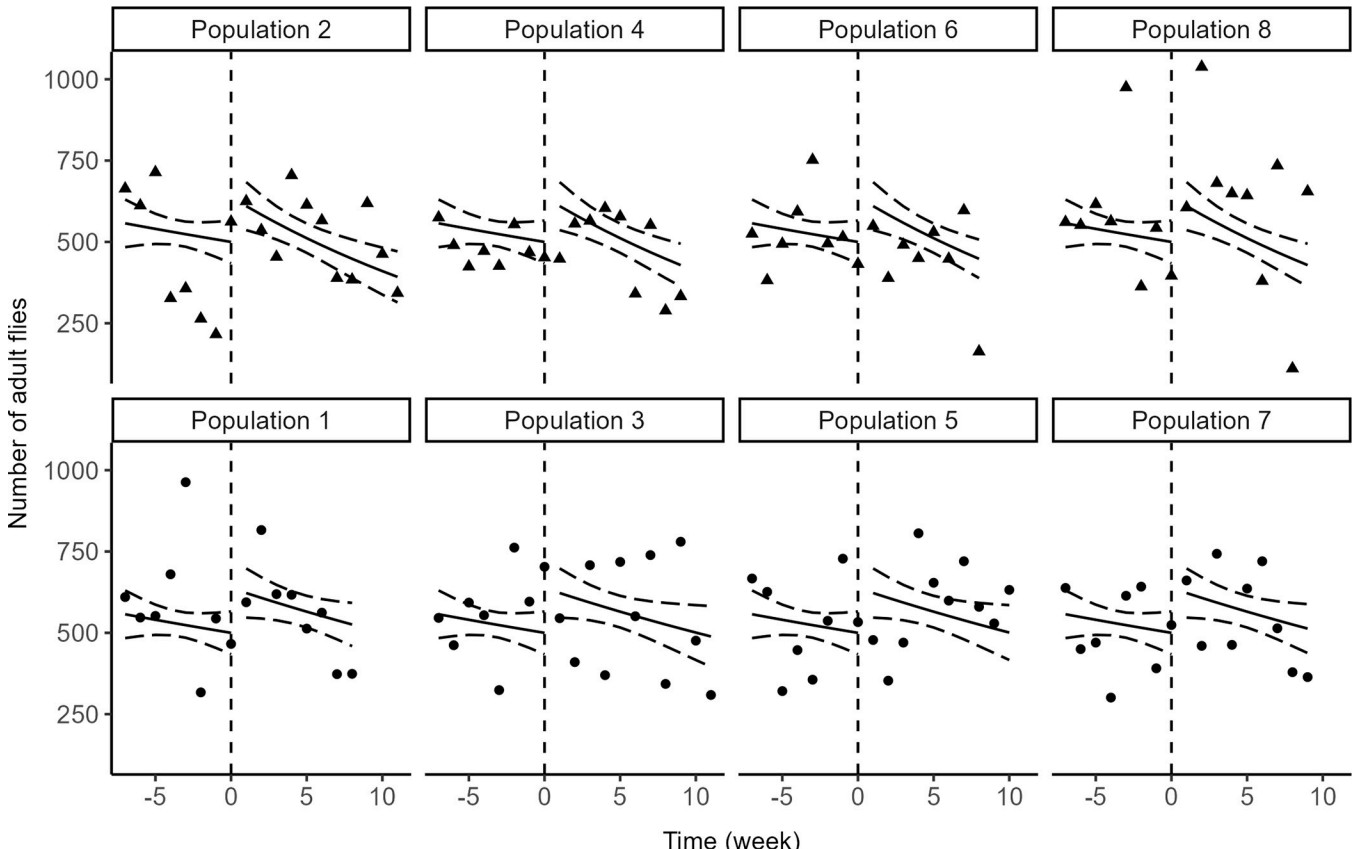

**Fig 5. Impacts of introduced CI-inducing flies.** Observed and predicted number of adult *D. suzukii* in populations with CI (triangles; introduction of individuals infected by *w*Tei) and without CI (circle; introduction of individuals infected by *w*Suz). Time is scaled so that introductions occur at *t* = 0.

growth rate. This basic idea has been formalized in the theoretical models of Dobson *et al.* [18] and Blackwood *et al.* [24] which predict that the introduction of one or more incompatible *Wolbachia* strains into a target population could lead to a transient decrease in population size [18]. Furthermore, assuming that a pre-existing Allee effect is enhanced by complementary tactics such as mating disruption, the unwanted population could be driven to extinction [24]. Although these models are of great interest for their efforts to link *Wolbachia* invasion to insect population dynamics, their hypotheses and predictions have not been tested with appropriate experiments. Using *Drosophila suzukii* as a model organism, we present here a first step in this direction.

Our main result is that introducing the incompatible *Wolbachia* strain *w*Tei into a recipient *Drosophila suzukii* population at carrying capacity does not result in an observable decrease in population size. This finding, with this specific *Wolbachia* strain and insect host, contradicts theoretical predictions [18,24], and moderates the perspective on using cytoplasmic incompatibility as a lever for insect pest management. In the remainder of the discussion, we explore the various hypotheses that could explain this result.

The most trivial hypothesis to explain the lack of influence of *Wolbachia*-induced CI on population dynamics is that contrary to expectations, the introduced incompatible *Wolbachia* did not invade the recipient populations and therefore, did not produce enough incompatible crosses to produce an effect. This is a hypothesis that we cannot firmly reject. Although the prevalence of individuals infected with *w*Tei reached up to 25% in some populations, weekly

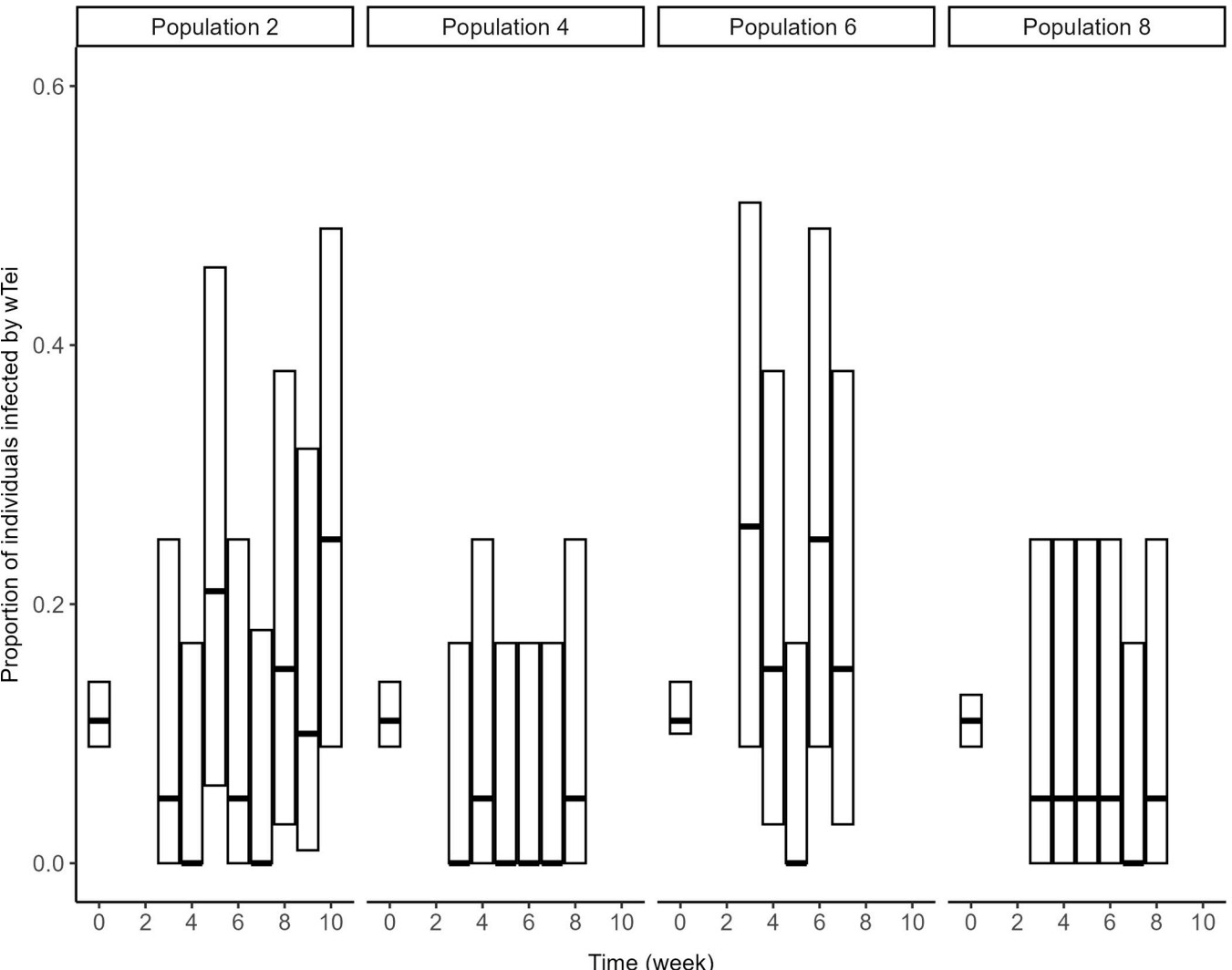

**Fig 6. Dynamics of *Wolbachia* infection.** Evolution of the proportion of *D. suzukii* infected by the introduced *Wolbachia* strain *w*Tei. Each proportion is presented with its theoretical 95% confidence interval, based on the binomial distribution. The first bar at *t* = 0 corresponds to the proportion of *w*Tei-infected flies actually introduced (10%). Subsequent prevalence was estimated from the third week post-introduction, after emergence of offspring from the introduced individuals.

estimates revealed important variations, with frequencies that were sometimes lower than the 10% fixed at introduction. This result contrasts with the published data, showing that incompatible *Wolbachia* strains introduced above their invasion threshold readily invade recipient populations. In the mosquito *Aedes aegypti*, the introduction of 20% of *Wolbachia*-infected females (with strain *w*AlbB) into an experimental population of uninfected individuals resulted in the fixation of the introduced strain in seven generations [46]. In another study on the same species, 65% of *w*Mel-infected mosquitoes were introduced into two populations of uninfected individuals, and again, the introduced strain became fixed 3 to 8 generations after introduction (30 to 80 days respectively [47]). In the fruit fly *Drosophila simulans*, caged populations were established with a mixture of uninfected individuals and either 35% of *w*Au-infected individuals or 4% of *w*Ri-infected individuals. A significant increase in *w*Au and *w*RI frequencies was observed in all populations from the 10[th] generation onwards, except in two

populations where *w*Ri was either lost or at low frequency [48]. With a similar invasion dynamic, our 4–5 generations experiment (seven to ten weeks) should have revealed the invasion of *w*Tei more clearly than it did. In this respect, our results are surprising.

Invasion may have failed if the number of infected drosophila introduced was too low to trigger invasion. We showed in the methods section that the absence of infection costs ($sf = 0$) suggests a theoretical invasion threshold close to 0% [9–13]. This means that in practice the introduction of very few individuals infected with *w*Tei should result in the invasion of this strain. Under these conditions, the number of flies we introduced experimentally—between 76 and 100 individuals in populations that were an order of magnitude larger—should have been high enough to induce invasion.

The observed frequencies of *w*Tei in the experimental populations with CI can alternatively be interpreted as a successful but slow invasion. In this context, the generational overlap inherent in our biological model could have allowed older males, inducing lower CI, to participate in reproduction. Indeed, CI intensity decreases rapidly with male age in several *Drosophila* species [49–51]. In *D. suzukii*, CI intensity is reduced to 27% in mass crosses between *w*Suz-infected females and *w*Tei-infected males aged 9–10 days, compared to 53% when males are 2–3 days old [32,34]. If older males are as competitive for access to females as younger ones, the negative relationship between CI intensity and male age could reduce the demographic consequences of CI, particularly in species such as *D. suzukii*, where individuals can survive for over 58 days at 23–24˚C [34].

A final hypothesis, not exclusive to the previous arguments, is that the influence of negative density dependence on host population dynamics overwhelms that of *Wolbachia*-induced CI. In our experiment, after an initial phase of exponential growth, the populations were well described by logistic-type models characterized by a carrying capacity estimated between 450 and 600 adults. This upper limit to population growth is probably a consequence of competition among larvae for limited food resources. Indeed, intraspecific competition certainly occurs in natural populations of *D. suzukii*. Our data suggest that the number of larvae developing in small fruits can be high (for instance, we could find up to 21 individuals at emergence from raspberry or blackberry exposed 48 h in the field; S1 Table). This suggests that larval intraspecific competition may be severe *in natura*, in particular when fruit abundance is weak, *i.e.*, at the beginning or at the end of the season. Laboratory experiments are consistent. According to Bezerra Da Silva *et al.* [36], the recurrent observation of *D. suzukii* pupae outside oviposition patches is a sign of high sensitivity to larval competition. The number of eggs and larvae developing in damaged fruits affects development time and further reduces adult size and fecundity [52,53]. In adults, increasing density results in females laying in fewer eggs laid per fruit, a behavior interpreted as an adaptation to alleviate competition among immature stages [54].

If larval survival depends on the intensity of competition within the fruit, a reduction in egg hatch rate resulting from *Wolbachia*-induced CI should, at high densities, alleviate competition and therefore, cause an immediate increase in larval survival. Reduced competition would offset incompatibility, and if this were true, CI would not cause the predicted decline in population size. Such a verbal argument echoes a specific section of the theoretical analysis of Dobson *et al.* ([18], Fig 2B), where the effect of density on immature survival is scaled by the parameter γ [55]. Increasing the influence of density by increasing γ made the effects of introducing incompatible strains disappear [18]. Other modelling studies have highlighted similar influence of competition among hosts on *Wolbachia* invasion [15–17]. More generally, the strength of competition may alter the benefits of gene drive [56] and of the sterile insect technique [57] with undesirable consequences such as increasing (rather than decreasing) the size of pest populations.

The extent to which negative density dependent survival may jeopardize attempts to reduce pest reproduction appears to be a key issue. However, empirical evidence for the critical role of density dependence in *Wolbachia* invasion and consequences for host population dynamics is completely lacking. In *D. suzukii*, we suggest that negative density dependence may explain the discrepancy between prior expectations and observed population dynamics. However, further species-specific models and experiments are needed to better understand the interplay between density dependence, the invasion of incompatible *Wolbachia* strains and the resulting population dynamics. Follow-up experiments could manipulate intraspecific competition to understand how competition interacts with CI. If there is an antagonistic interaction between the two processes, the population consequences of CI should only appear when flies are released before the populations reach their carrying capacity.

Positive density dependence, *i.e.*, the Allee effect, is also being proposed as an important process in pest management. The basic reasoning is that eradication can be achieved *via* a combination of treatments that reduce population size (pesticides, predators, etc.) and tactics that enhance a pre-existing Allee effect (for instance, mating disruption). Theoretical models, verbal arguments and reviews are numerous [19,20,58,59] and, for insect pests, well supported by widespread evidence of Allee effects [21,60]. A keystone example is that of the gypsy moth *Lymantria dispar*, whose spread is slowed by a combination of insecticides, sterile insect technique and mating disruption [61]. From this perspective, an important finding of our study is the absence of a demographic Allee effect in *D. suzukii*. Despite founding populations with the smallest possible number of individuals (one male and one female) we observed no positive relationship between population size and growth rate, and no extinction. This finding contrasts with observations of Allee effects in closely related species such as *D. melanogaster* [37,62]. In our experimental conditions with population cages, relatively high densities may have prevented an Allee effect. Alternatively, a component Allee effect could have occurred at the individual level, but was offset by negative density dependence and thus did not translate to the population level.

## Conclusion

In conclusion, many of our results are puzzling with respect to prior expectations. The absence of a transient decrease in *D. suzukii* population size after the introduction of the incompatible *Wolbachia* strain *w*Tei is most easily explained by the slow invasion of this introduced strain, which, in itself, lacks clear explanations. Nevertheless, there were some incompatible crosses in populations where the proportion of *w*Tei was sometimes higher than 20%, but these did not influence the host population dynamics. We therefore conclude that the strong density dependent competition could be a key driver of insect population dynamics, counterbalancing the predicted effect of cytoplasmic incompatibility, and possibly hampering pest management programs based on reproductive failure. *Wolbachia*-based methods, just like any other methods that reduce pest realized fecundity, should be implemented when populations are not under an overwhelming competition, for instance, at the beginning of the season.

## Supporting information

**S1 Fig. Different areas used for the census method.** Black square represents area boundaries. Green and red arrows indicate which flies are counted and which are not. Area 1: flies appearing on the front and on the rear panels, Area 2: individuals appearing on the whole surface except the left side and the bottom of the cage and Area 3: only individuals appearing on the front panel.
(DOCX)

**S2 Fig. Melting curves analysis of DNA fragments after PCR for different *Wolbachia* strains in *Drosophila suzukii*.** Here, are plotted the negative first derivative of the fluorescence (-d(RFU)/dt $(10^3)$ *versus* temperature, peaks corresponding to the melting temperature Tm of *Wolbachia* DNA (Blue line: *w*Suz, green line: *w*Tei, red line: *w*Ha present in third transinfected line of *Drosophila suzukii*, which was not used for these experiments). FRU means relative fluorescence units.
(DOCX)

**S3 Fig. Schematic representation of the statistical model fitted to time variations in number of *D. suzukii* in order to test the effect of introducing insects infected with either incompatible (*w*Tei) or compatible (*w*Suz) *Wolbachia* into a recipient population.** Four hypotheses were tested, based on estimation of four model parameters. H1: parameter $\beta_1$ population growth before introduction; populations were at carrying capacity and we therefore expected $\beta_1 = 0$; H2: parameter $\beta_2$ represents the immediate effect of introduction on population size; the number of flies introduced was set at 10% of the carrying capacity so we expected $\beta_2 > 0$. H3: parameter $\beta_3$ represents the effect of introduction on time variation of population size; if $\beta_1 = 0$ and $\beta_2 > 0$, we expected $\beta_3 < 0$ reflecting a return to carrying capacity. H4: parameter $\beta_4$ represents the effect of the *Wolbachia* strain introduced on the variation of population size after introduction; theoretical models predict that the introduction of an incompatible strain (*w*Tei) should result in a transient decrease in the population size, which should not be observed in control populations (introduction of *w*Suz), so that we expect $\beta_4 < 0$.
(DOCX)

**S1 Table. Number of adults *D. suzukii* emerging from fruits exposed in the field.**
(DOCX)

## Acknowledgments

We thank P. Decoeur, E. Deleury, E. Desouhant, P. Gibert, C. Lopes, L. van Oudenhove for useful discussion on data.

## Author Contributions

**Conceptualization:** Alexandra Auguste, Nicolas Ris, Laurence Mouton, Xavier Fauvergue.

**Formal analysis:** Alexandra Auguste.

**Investigation:** Alexandra Auguste, Zainab Belgaidi, Laurent Kremmer.

**Methodology:** Alexandra Auguste, Nicolas Ris, Laurence Mouton, Xavier Fauvergue.

**Project administration:** Laurence Mouton, Xavier Fauvergue.

**Supervision:** Xavier Fauvergue.

**Visualization:** Alexandra Auguste.

**Writing – original draft:** Alexandra Auguste.

**Writing – review & editing:** Nicolas Ris, Laurence Mouton, Xavier Fauvergue.

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
