## [Decision Letter · Decision Letter 0]

30 Oct 2023

PONE-D-23-25628Insect population dynamics under *Wolbachia*-induced cytoplasmic incompatibility: puzzle more than buzz in *Drosophila suzukii*PLOS ONE

Dear Dr. Auguste,

Thank you for submitting your manuscript to PLOS ONE. After careful consideration, we feel that it has merit but does not fully meet PLOS ONE’s publication criteria as it currently stands. Therefore, we invite you to submit a revised version of the manuscript that addresses the points raised during the review process.

ACADEMIC EDITOR: Revision of the article is required for the improvement of the draft. Some suggested changes are in comments portion to revise the manuscript. Please find the comments and suggested corrections. Complete editing corrections, journal-style format, use of abbreviation, missing information, use of abbreviations and reference writing should be carefully revised keeping in view the author's instructions. Please submit your revised manuscript by Dec 14 2023 11:59PM. If you will need more time than this to complete your revisions, please reply to this message or contact the journal office at plosone@plos.org. Please include the following items when submitting your revised manuscript:A rebuttal letter that responds to each point raised by the academic editor and reviewer(s). You should upload this letter as a separate file labeled 'Response to Reviewers'.A marked-up copy of your manuscript that highlights changes made to the original version. You should upload this as a separate file labeled 'Revised Manuscript with Track Changes'.An unmarked version of your revised paper without tracked changes. You should upload this as a separate file labeled 'Manuscript'.

We look forward to receiving your revised manuscript.

Kind regards,

Bilal Rasool, PhD

Academic Editor

PLOS ONE

Reviewers' comments:

Reviewer's Responses to Questions

**Comments to the Author**

1. Is the manuscript technically sound, and do the data support the conclusions?

Reviewer #1: Yes

Reviewer #2: Partly

Reviewer #3: Yes

2. Has the statistical analysis been performed appropriately and rigorously? 

Reviewer #1: Yes

Reviewer #2: Yes

Reviewer #3: Yes

3. Have the authors made all data underlying the findings in their manuscript fully available?

Reviewer #1: Yes

Reviewer #2: No

Reviewer #3: Yes

4. Is the manuscript presented in an intelligible fashion and written in standard English?

Reviewer #1: Yes

Reviewer #2: Yes

Reviewer #3: Yes

5. Review Comments to the Author

Reviewer #1: The manuscript entitled “Insect population dynamics under Wolbachia-induced cytoplasmic incompatibility: puzzle more than buzz in Drosophila suzukii” describes the efforts of the authors to confirm the models of Dobson and Blackwood in a laboratory setup. The authors used a well-known Wolbachia infected strain wTei which has also been introduced in the pest insect Drosophila suzukii and tested whether the introduction of wTei infected individuals would result in invasion of the Wolbachia in the population and subsequent population decline. Their results are quite surprising since they did not observe decrease in the population size of D. suzukii over time. They also did not identify any Allee effect but only negative density dependence. Their results can be used in the planning of CI strategies aiming to reduce the population of pest insects. In general, this a well-written manuscript with sound scientific evidence and I recommend it for publishing in Plos One.

Below are some minor comments that will further improve this manuscript:

L16-23: this information is too general for the abstract. It is better to be moved to the introduction and reform the abstract in giving the most important piece of information about this study.

L42: “viable” offspring

L49: “has” piqued

L103: delete “The present article reports our first findings.”

L104-106: delete all the different types of fruits

L117: replace “produce” with “cause”

L129-130:this part belongs to the discussion

L148: please define what is the “black wall”. There is no black wall in the photo of the supplementary material

L151: please explain why you decided not to use area No3 in this study

L155: add the reference of the R software

L161-162: this part is not clear to me. Please explain how you handled the pairs of D. suzukii step by step to make it clearer to the readers who will wish to replicate it.

L203: give the full name of the AIC metric

L265: this figure should be moved to the supplementary material

L297: this is Figure 3, and not 1

Reviewer #2: Article PONE-D-23-25628 lacks some important details to be fully understood for the readers. Please take into account that many sentences need to be rephrased. Some suggested changes as an example are in the comments portion to revise and improve the manuscript. There are many sentences throughout the manuscript which is hard to understand. The present form of draft required a lot of corrections. Please find the comments and suggested corrections. Complete editing corrections, journal-style format, use of abbreviation and missing information should be maintained.

Title: Revise the title of the paper

Abstract: Revision of the abstract is required.

Line 19-20: First, when an incompatible Wolbachia strain….. triggering Wolbachia invasion” Rephrase the sentence

Line 26: exogenous” please change the word with suitable word

Line 29: From these results, we propose that negative density dependence is an important but underappreciated…”Restructure the sentence

Rewrite the abstract portion as this is not understandable. The English writing is so confusing to read, it is therefore suggested to rewrite the abstract.

Line 32: Early-season populations?

Introduction:

Line 40-41: CI is the consequence….infected male. Restructure the sentence and try to avoid abbreviation starting the sentence

Line 48-49: The high invasion success of Wolbachia…. as piqued the interest of epidemiologists” Rephrase the sentence. Replace piqued with suitable alternative word.

Line 104-106: There are several reasons…..etc… “Restructure the sentence as this is lengthy and also follow the author’s instructions while writing the text.

Line 137: females[32,34] please follow the space, commas, abbriviations etc instructions while writing text” this may be rechecked throughout the manuscript

Line 139: climatic chamber” manufacturer details

Line 137-140: Reference?

Line 143: photographed cages?

Line 181-185: Nt = N0 exp( r t ) follow the authors instructions while writing the formula or equation. mention the reference of the formulas used in the draft

Line 199: (1-49, 50-99, …, 850-899)? Follow the authors instructions while writing the text

Line 190-204: Mention the references of the formulas/equations used in the draft

Line 203-204: AIC? What is this abbreviation” please mention where firstly used

Line 220: "IC" populations.?

Line 223: CI populations?

Line 227-229: 81f” 81 F “Braig et al, 1998” mention reference number and also include in reference list” Please also write the reference number at appropriate place. Are these 2R or 691 R primers?. Please recheck?

Line 230-233: Please add reference

Line 235-242: Add reference

Line 252: please add reference of the formula, equations used

Line 257-264: Please recheck for if there are any errors and correct if necessary

Line 295: “• “Please recheck what is this

Line 303: spacing

Table 1: AIC and RMSE? Abbreviations

Figure 4: Replace the legend as “Proportion of the populations increasing (t to t+1)”

Please recheck the figure legends and change if necessary week or weeks

Line 385-387: Please add suitable reference

Line 492-499: Please add precise suitable conclusion with future implications of the study

References: Follow the journal style formatting? Please add the references which are mentioned in the text and not present in the reference list.

Please double-check for typos and inconsistencies in Journal style/formatting, missing italics, missing information etc

Reviewer #3: The study by Auguste et al. fills an important gap in our knowledge of bacterial symbionts and their potential application in pest control, as it is one of the few studies that has attempted to test theoretical models under rigorous experimental conditions. In this case, the study aims at testing two models, those of Dobson et al. 2002 and of Blackwood et al. 2018, that addressed the effect of Wolbachia-induced cytoplasmic incompatibility on population dynamics of pests, ultimately and potentially leading to their extinction, with or without a preexisting Allee effect, and the potential use of CI Wolbachia as pest control method. The results presented are surprising and at times puzzling, as none of the major theoretical predictions seems to hold true. This study represents an important contribution to the field and is likely to generate discussion and in turn lead to more (and much needed!) experimental studies.

I have no major concerns regarding the experimental conditions, but some things need to be clarified, and the discussion expanded to some extent.

- Authors should elaborate more on how the invasion threshold was established and the subsequent choice of introducing 10% of Wolbachia transinfected flies, and why they did not test also introducing more Wolbachia transinfected flies. Please do the math for the readers. This should be explained in the methods section, as now the 10% figure at line 221 appears without prior explanation.

- Similarly, when comparing with invasion thresholds used in other studies (lines 412-420), these should be presented in a way that makes comparisons more immediate. Again, please do the math for the readers.

- Authors should also elaborate on parameters that could be modified in potential follow up experiments. For example, I think it could be interesting to discuss the role that the carrying capacity (a debated concept to begin with) model parameter may have played in the differences between theory and experiments. Perhaps flies should be released before population reaches carrying capacity when negative density dependence is to be expected like in D. suzukii? Could this be a follow up experiment?

Additional minor comments:

- Line 183: what are missing data in this case? Please specify.

- Lines 216-217: unless authors think that the interruption of the experiments changed the trajectory of the experiment, I’d remove this sentence. Otherwise, the impact of this interruption should be discussed.

- HRM conditions should be better described (e.g., temperature ramp rate) (lines 231-233). Authors should also include (in the supplementary material) an image of the HRM melting curves.

- Please carefully check spelling throughout the manuscript.

6. PLOS authors have the option to publish the peer review history of their article (what does this mean?). If published, this will include your full peer review and any attached files.

Reviewer #1: No

Reviewer #2: No

Reviewer #3: No

---

## [Author Response · Author response to Decision Letter 0]

31 Jan 2024

Response to Reviewers

Dear Editors and Reviewers, we thank you for your valuable comments on our manuscript “Insect population dynamics under Wolbachia-induced cytoplasmic incompatibility: puzzle more than buzz in Drosophila suzukii” and for giving us the opportunity to submit a revised version. We have considered all these comments in preparing the revised manuscript. Below is a concise description of how we addressed each comment.

Reviewer #1

The manuscript entitled “Insect population dynamics under Wolbachia-induced cytoplasmic incompatibility: puzzle more than buzz in Drosophila suzukii” describes the efforts of the authors to confirm the models of Dobson and Blackwood in a laboratory setup. The authors used a well-known Wolbachia infected strain wTei which has also been introduced in the pest insect Drosophila suzukii and tested whether the introduction of wTei infected individuals would result in invasion of the Wolbachia in the population and subsequent population decline. Their results are quite surprising since they did not observe decrease in the population size of D. suzukii over time. They also did not identify any Allee effect but only negative density dependence. Their results can be used in the planning of CI strategies aiming to reduce the population of pest insects. In general, this a well-written manuscript with sound scientific evidence and I recommend it for publishing in PlosOne.

We thank Reviewer #1 for this positive general appreciation.

Below are some minor comments that will further improve this manuscript:

L16-23: this information is too general for the abstract. It is better to be moved to the introduction and reform the abstract in giving the most important piece of information about this study.

Ok. We removed the three first sentences, as suggested. The abstract now goes straight to the point. These changes on the abstract also address the second comment of Reviewer #2. We indeed revised the abstract thoroughly.

L42: “viable” offspring

Ok, we have added “viable” offspring as suggested (L37).

L49: “has” piqued.

Ok, we have indeed replaced “as piqued” by “has aroused” in order to address another comment from Reviewer #2 on this point (L42).

L103: delete “The present article reports our first findings.”

We agree with the reviewer that this sentence could be considered out of place and we have removed it.

L104-106: delete all the different types of fruits.

Ok, as suggested, we have removed the list of types of fruits that can be infested by Drosophila suzukii. The sentence now reads much better (L92-93):

“First, D. suzukii is a major invasive pest of berry and stone fruit crops [25]”.

L117: replace “produce” with “cause”

Ok, we have replaced “produce” with “cause” (L103).

L129-130: this part belongs to the discussion

The last sentence of the introduction was meant as an ouverture to the discussion. It could be removed. However, placed here at the end of the Introduction, it reveals a major highlight of our study and should therefore add to the excitement of reading the following two sections. Nevertheless, to decrease the feeling that this sentence is part of the discussion we have replaced "hypothesize" with "propose” in order. This now gives (L112-115):

“We discuss these results in the light of the models developed by Dobson et al. [18] and Blackwood et al. [24]. and propose that density dependent competition may have much stronger influences than expected on Wolbachia invasion and the demographic consequences of cytoplasmic incompatibilities.”

L148: please define what is the “black wall”. There is no black wall in the photo of the supplementary material

We indeed wrote “back wall”, not “black wall”. But we take this comment as an indication that the term could be misleading so we have replaced “back wall” with “rear panel” (L132).

L151: please explain why you decided not to use area No3 in this study.

Thank you for pointing this, as it was not clear in the first submission. We now thoroughly justify the use of areas 1 and 2, not area 3. For this, we have added a few sentences in the Result section (L269-274):

“Given the high coefficients of determination obtained for areas 1 and 2 and the small number of populations involved, these two areas were chosen to estimate the population size in experiments 2 and 3 respectively. However, should the number of experimental populations increase, we would recommend the use of area 3, because counting higher numbers of insects in the other two areas would be too time consuming to characterize many populations on a weekly basis. Therefore, area 3 was not used in the present study, but it will be used in the future.”

We nevertheless wish to keep our results on area 3 in order to use this first article as a reference on methodology in future articles. This is stated in the M&M (L134-136):

“Areas 1 and 2 were used for the experiments 2 and 3, respectively. Area 3 was not used in the present study but it will be used in follow-up articles. The reasons for this choice are given in the Results section”.

L155: add the reference of the R software.

Ok. We have added the following reference in the text and bibliography (L140): R Core Team. R: A Language and Environment for Statistical Computing [Internet]. Vienna, Austria: R Foundation for Statistical Computing; 2022. Available from: https://www.R-project.org/

L161-162: this part is not clear to me. Please explain how you handled the pairs of D. suzukii step by step to make it clearer to the readers who will wish to replicate it.

Thank you for this comment; we understand that our writing was here too concise to be fully understandable. We have tried to clarify how we handled male-female pairs to initiate the Drosophila suzukii populations in Experiment 2. The new text is now (L144-149): 

“To this end, we founded D. suzukii populations with different numbers of male-female pairs and estimated subsequent growth rates. Founding pairs were formed with 1 male and 1 female, all virgins and less than 24 hours old. Populations were founded with either 1, 3, 10, 30 or 100 pairs, and each of these initial population size was replicated between 8 and 13 times (13 populations with 1 pair, 13 with 3 pairs, 10 with 10 pairs, 8 with 30 pairs, and 8 with 100 pairs, for a total of 52 populations). Replicated populations were distributed across four blocks spaced one week apart.”

L203: give the full name of the AIC metric

Ok, we have given the full name of AIC, that is, Akaike Information Criterion (L194). 

L265: this figure should be moved to the supplementary material

Figure 1 was intended to provide a quicker understanding of the complex statistical models and working hypotheses underlying the data analyses and interpretations, but we recognize that while it provides a visual summary, it is partially redundant with the text. In response to Reviewer #1, this figure has been moved to the Supplementary Material.

L297: this is Figure 3, and not 1

Yes! Thanks for the comment, we have corrected this error (L297).

Reviewer #2

Article PONE-D-23-25628 lacks some important details to be fully understood for the readers. Please take into account that many sentences need to be rephrased. Some suggested changes as an example are in the comments portion to revise and improve the manuscript. There are many sentences throughout the manuscript which is hard to understand. The present form of draft required a lot of corrections. Please find the comments and suggested corrections. Complete editing corrections, journal-style format, use of abbreviation and missing information should be maintained.

Title: Revise the title of the paper

Contrary to most of the other comments, we would like to argue on this specific point: we think that this title closely reflects our approach (insect population dynamics under the influence of CI induced by Wolbachia) as well as our surprising results (as pointed out by Reviewer #1 and Reviewer #2) - surprising and therefore puzzling given the model predictions. Reviewer #3 even used the term “puzzling” in his main comment below. We also think that this is a very attractive title for potential readers, which should contribute to article visibility and future citations. Therefore, although Reviewer #2 has specific arguments against this title that we did not understand yet, we would be grateful if it could remain as it is.

Abstract: Revision of the abstract is required.

Okay. We have revised the abstract according to Reviewer #1's comments, removing some contextual information that might seem too general for an abstract. We have also reworked this entire section and hope that the new version is better. The abstract is now (L15-27):

“In theory, the introduction of individuals infected with an incompatible strain of Wolbachia pipientis into a recipient host population should result in the symbiont invasion and reproductive failures caused by cytoplasmic incompatibility (CI). Modelling studies combining Wolbachia invasion and host population dynamics show that these two processes could interact to cause a transient population decline and, in some conditions, extinction. However, these effects could be sensitive to density dependence, with the Allee effect increasing the probability of extinction, and competition reducing the demographic impact of CI. We tested these predictions with laboratory experiments in the fruit fly Drosophila suzukii and the transinfected Wolbachia strain wTei. Surprisingly, the introduction of wTei into D. suzukii populations at carrying capacity did not result in the expected wTei invasion and transient population decline. In parallel, we found no Allee effect but strong negative density dependence. From these results, we propose that competition interacts in an antagonistic way with Wolbachia-induced cytoplasmic incompatibility on insect population dynamics. If future models and data support this hypothesis, pest management strategies using Wolbachia-induced CI should target populations with negligible competition but a potential Allee effect, for instance at the beginning of the reproductive season.”

Line 19-20: First, when an incompatible Wolbachia strain….. triggering Wolbachia invasion” Rephrase the sentence

Yes, please see above. This sentence has been changed according to comments of Reviewer #1.

Line 26: exogenous” please change the word with suitable word

Ok, we have replaced “exogeneous” with “transinfected” (L21).

Line 29: From these results, we propose that negative density dependence is an important but underappreciated…”Restructure the sentence

As suggested, we have restructured the sentence (L24-25):

“From these results, we propose that competition interacts in an antagonistic way with Wolbachia-induced cytoplasmic incompatibility on insect population dynamics.”

Rewrite the abstract portion as this is not understandable. The English writing is so confusing to read, it is therefore suggested to rewrite the abstract.

As mentioned above, we indeed rewrote the abstract entirely and hope that this section is not confusing anymore.

Line 32: Early-season populations?

Thanks for the comment, this term could be fuzzy. We have corrected the sentence (L25-27):

“If future models and data support this hypothesis, pest management strategies using Wolbachia-induced CI should target populations with negligible competition but a potential Allee effect, for instance, at the beginning of the reproductive season”. 

Introduction: 

Line 40-41: CI is the consequence….infected male. Restructure the sentence and try to avoid abbreviation starting the sentence

Ok, we have restructured the sentence and removed abbreviation (L34-37):

“Cytoplasmic incompatibility (CI) is one such manipulation that occurs, in its simplest form, when infected males cross with uninfected females. Wolbachia-induced sperm modification in males cannot be restored by females. In diploid species, these crosses result in embryo death while others produce viable offspring”.

Line 48-49: The high invasion success of Wolbachia…. as piqued the interest of epidemiologists” Rephrase the sentence. Replace piqued with suitable alternative word.

The sentence was corrected and “picked” was replaced with “aroused”, as suggested by Reviewer #1. This now gives (L42):

“The high invasion success of Wolbachia combined with its ability to reduce the transmission of viruses or other pathogens [6,7] has aroused the interest of epidemiologists.”

Line 104-106: There are several reasons…..etc… “Restructure the sentence as this is lengthy and also follow the author’s instructions while writing the text.

Referee #1 has also suggested to shorten the sentence. We have therefore removed the list of fruits possibly infected by D. suzukii, which now gives (L92-93):

“First, D. suzukii is a major invasive pest of berry and stone fruit crops [25].”

Line 137: females[32,34] please follow the space, commas, abbriviations etc instructions while writing text” this may be rechecked throughout the manuscript

Ok, and thank you for pointing this type. We have added a space here, but more generally, we have edited the entire manuscript to better fit to Plos One instructions for authors.

Line 139: climatic chamber” manufacturer details

Our climate chambers are an integral part of the building; they were built with the building. They are not specially branded. Could we refer to them as built-in climate chambers? If yes this would now give (L124):

“Rearing, as well as all experiments, occurred in a built-in climatic chamber at 23-24°C, 50-60% humidity and a photoperiod of 12 h-12 h.”

Line 137-140: Reference?

The rearing medium that we describe in extenso was home-made, but we have nonetheless added a reference for the first article that described it (Iacovone et al. 2018) (L122).

Line 143: photographed cages?

We have replaced “photographed cages” with “photo-based census method” (L127)

Line 181-185: Nt = N0 exp( r t ) follow the authors instructions while writing the formula or equation. mention the reference of the formulas used in the draft.

This equation is the most basic model of exponential growth that now belongs to the “general knowledge” and is generally cited to as such, without reference. Should we add a reference, it would be Malthus 1798. 

Furthermore, we have added a reference number to each equation of the manuscript.

Line 199: (1-49, 50-99, …, 850-899)? Follow the authors instructions while writing the text

Ok, we have replaced “(1-49, 50-99, …, 850-899)” with “(1-49, 50-99, etc.)” (L186). 

Line 190-204: Mention the references of the formulas/equations used in the draft

As mentioned above, the equations were not actually referenced in the submitted manuscript. We have now added a reference to each equation.

Line 203-204: AIC? What is this abbreviation” please mention where firstly used

As also suggested by Reviewer #1, we have added the full name of AIC (Akaike Information Criterion) (L194).

Line 220: "IC" populations.?

Thanks for the comment. This typo has been corrected and we have now explained what are “control” populations and “CI” populations. This now gives (L208-210):

“Individuals introduced after the seven-week period of dynamic equilibrium were infected by either wSuz for the four "control" populations (without cytoplasmic incompatibility) or wTei for the four "CI" populations (with cytoplasmic incompatibility)”.

Line 223: CI populations?

Yes indeed, CI for Cytoplasmic Incompatibility, as mentioned above.

Line 227-229: 81f” 81 F “Braig et al, 1998” mention reference number and also include in reference list” Please also write the reference number at appropriate place. Are these 2R or 691 R primers?. Please recheck?

Ok, thanks a lot. We have added the missing reference to the bibliography (L226): Braig HR, Zhou W, Dobson SL, O’Neill SL. Cloning and characterization of a gene encoding the major surface protein of the bacterial endosymbiont Wolbachia pipientis. Journal of Bacteriology 1998;180(9):2373‑8. 

We have also rechecked the name of primers. There is no error: 2R primers have indeed been used routinely to characterize Wolbachia infection status in many arthropods.

Line 230-233: Please add reference

Ok, we have added the appropriate reference (L229): Henri H, Mouton L. High‐Resolution Melting technology: a new tool for studying the Wolbachia endosymbiont diversity in the field. Molecular Ecology Resources. 2012;12(1):75‑81. 

Line 235-242: Add reference

Ok, we have added reference (L236): Archontoulis SV, Miguez FE. Nonlinear regression models and applications in agricultural research. Agronomy Journal. 2015;107(2):786‑98.

Line 252: please add reference of the formula, equations used

Please see above, this is a generalized linear statistical model that has no specific reference in the literature. An equation number was however added, similarly to all equations in the text.

Line 257-264: Please recheck for if there are any errors and correct if necessary

Thank you for your scrutiny. We indeed double-checked and found no error in the description of these four hypotheses.

Line 295: “• “Please recheck what is this

Ok, we removed such dots. Dots are sometimes used as multiplication sign. 

However, for simplicity and consistency, we replaced all these dots with spaces in equations over the whole manuscript. This does appear as a common practice in other manuscript published in Plos One.

Line 303: spacing

Ok, we assume that you are commenting on spaces between operators (such as the equal operator) and figures or parameters. For consistency over the whole manuscript, we have added spaces before and after each operator.

Table 1: AIC and RMSE? Abbreviations

Ok, and as mentioned above, we have added full names for AIC and RMSE in the legend of the table 1. RMSE stands for root mean square error (L316).

Figure 4: Replace the legend as “Proportion of the populations increasing (t to t+1)”

Ok, we have replaced the legend of the Fig 4, which is now Fig 3, as suggested (L297).

Please recheck the figure legends and change if necessary week or weeks

Units of time (generation, week, day in the axis title) are used indifferently in singular or plural in other published articles. For the sake of consistency, we have decided to use the singular for each of the figure legends.

Line 385-387: Please add suitable reference

Ok, we have added reference (L369): Werren JH, Baldo L, Clark ME. Wolbachia: master manipulators of invertebrate biology. Nature Reviews Microbiology. 2008;6(10):741‑51.

Line 492-499: Please add precise suitable conclusion with future implications of the study

Reviewer #2 is right; the end of the manuscript could be improved with a clear conclusion highlighting future implications of our study.

We have created a separate section entitled Conclusions at the end of the discussion (L464-474):

"In conclusion, many of our results are puzzling with respect to prior expectations. The absence of a transient decrease in D. suzukii population size after the introduction of the incompatible Wolbachia strain wTei is most easily explained by the slow invasion of this introduced strain, which, in itself, lacks clear explanations. Nevertheless, there were some incompatible crosses in populations where the proportion of wTei was sometimes higher than 20%, but these did not influence the host population dynamics. We therefore conclude that the strong density dependent competition could be a key driver of insect population dynamics, counterbalancing the predicted effect of cytoplasmic incompatibility, and possibly hampering pest management programs based on reproductive failure. Wolbachia-based methods, just like any other methods that reduce pest realized fecundity, should be implemented when populations are not under an overwhelming influence of competition, for instance, at the beginning of the season.”

References: Follow the journal style formatting? Please add the references which are mentioned in the text and not present in the reference list.

Thanks for the comments. One reference was indeed missing from the bibliography (L226): Braig HR, Zhou W, Dobson SL, O’Neill SL. Cloning and characterization of a gene encoding the major surface protein of the bacterial endosymbiont Wolbachia pipientis. Journal of Bacteriology. 1998;180(9):2373‑8. 

According to Reviewer #2, we have checked again that all references mentioned in the text are present in the reference list.

Please double-check for typos and inconsistencies in Journal style/formatting, missing italics, missing information etc

Thanks for the comment. As also suggested by Reviewer #1, we have double- checked the manuscript for typos (missing italics, spacing between operator, reference of equation, etc.)

Reviewer #3

The study by Auguste et al. fills an important gap in our knowledge of bacterial symbionts and their potential application in pest control, as it is one of the few studies that has attempted to test theoretical models under rigorous experimental conditions. In this case, the study aims at testing two models, those of Dobson et al. 2002 and of Blackwood et al. 2018, that addressed the effect of Wolbachia-induced cytoplasmic incompatibility on population dynamics of pests, ultimately and potentially leading to their extinction, with or without a preexisting Allee effect, and the potential use of CI Wolbachia as pest control method. The results presented are surprising and at times puzzling, as none of the major theoretical predictions seems to hold true. This study represents an important contribution to the field and is likely to generate discussion and in turn lead to more (and much needed!) experimental studies.

We thank reviewer #3 for this generally positive comment. Indeed, this manuscript is the first of a series that will document follow-up research aimed at deciphering the puzzling results presented here. As a teaser, we can reveal here that future experimental approaches will confirm our hypothesis that density-dependent competition may doom pest management strategies based on IIT and SIT.

I have no major concerns regarding the experimental conditions, but some things need to be clarified, and the discussion expanded to some extent.

Authors should elaborate more on how the invasion threshold was established and the subsequent choice of introducing 10% of Wolbachia transinfected flies, and why they did not test also introducing more Wolbachia transinfected flies. Please do the math for the readers. This should be explained in the methods section, as now the 10% figure at line 221 appears without prior explanation.

This is a crucial point and we thank Reviewer #3 for raising it. In the first submission, the 10% invasion threshold came without justification in the Method section. We had explained its implementation later in the discussion, but evidently, this was too late for a scientist reading linearly. We therefore changed our text to justify the 10% earlier in the Methods section. We now write (L210-219):

“Theoretically, Wolbachia invasion depends on the intensity of CI, transmission efficiency, and infection cost [9–13]. Assuming perfect transmission, an incompatible strain should invade if it is introduced above a threshold equal to the ratio of infection cost (parameter sf in the models referenced above) to incompatibility (parameter sh). In our system of D. suzukii and wTei, incompatibility is incomplete, with an egg hatch rate of about 33% after incompatible crosses (i.e., sh = 0.67; [34]). In parallel, the analyses of traits such as fecundity, longevity, egg hatch rate and developmental time all suggest an absence of cost of wTei infection on D. suzukii [32,34]. This apparent absence of infection costs (sf = 0), even with incomplete incompatibility, suggests a theoretical invasion threshold of 0%. In practice, this would mean just enough individuals to compensate for the stochastic events that are not considered in theoretical models. A number of individuals close to 10% of the mean number of individuals over the equilibrium period should be sufficient to trigger invasion.”

- Similarly, when comparing with invasion thresholds used in other studies (lines 412-420), these should be presented in a way that makes comparisons more immediate. Again, please do the math for the readers.

Thank you again for this interesting comment. It is true that results from different experiments are uneasy to compare. For this reason, we have now tuned the various published results to common metrics. This now gives (L391-402):

“This result contrasts with the published data showing that incompatible Wolbachia strains introduced above their invasion threshold readily invade recipient populations. In the mosquito Aedes aegypti, the introduction of 20% of Wolbachia-infected females (with strain wAlbB) into an experimental population of uninfected individuals resulted in the fixation of the introduced strain in seven generations [46]. In another study on the same species, 65% of wMel-infected mosquitoes were introduced into populations of uninfected individuals, and again, the introduced strain became fixed 3 to 8 generations after introduction (30 to 80 days respectively [47]). In the fruit fly Drosophila simulans, caged populations were established with a mixture of uninfected individuals and either 35% of wAu-infected individuals or 4% of wRi-infected individuals. A significant increase in wAu and wRI frequencies was observed in all populations from the 10th generation onwards, except in two populations where wRi was either lost or at low frequency [48]. With a similar invasion dynamic, our 4-5 generation experiment (seven to ten weeks) should have revealed the invasion of wTei more clearly than it did. In this respect, our results are surprising.”

- Authors should also elaborate on parameters that could be modified in potential follow up experiments. For example, I think it could be interesting to discuss the role that the carrying capacity (a debated concept to begin with) model parameter may have played in the differences between theory and experiments. Perhaps flies should be released before population reaches carrying capacity when negative density dependence is to be expected like in D. suzukii? Could this be a follow up experiment?

This is again a very interesting idea. Follow-up experiments were designed to test the effect of intraspecific competition on CI outcomes and the effect of Wolbachia introduction in growing populations.

We have added the following sentences in the Discussion (L447-450): “Follow-up experiments could manipulate intraspecific competition to understand how competition interacts with CI. If there is an antagonistic interaction between the two processes, the population consequences of CI level should only appear when flies are released before the populations reach their carrying capacity. “

Additional minor comments:

- Line 183: what are missing data in this case? Please specify

The comment is worth as we did not explain why and which data were removed. We now explain this better. Now we write (L172-175):

“The number of Individuals estimated immediately after release (week 1) was missing for all populations from the first block. It was therefore impossible to add the initial number as an offset to the model. Therefore, the five populations of the first block were discarded due to missing values.”

- Lines 216-217: unless authors think that the interruption of the experiments changed the trajectory of the experiment, I’d remove this sentence. Otherwise, the impact of this interruption should be discussed.

Ok. We have removed this sentence. 

- HRM conditions should be better described (e.g., temperature ramp rate) (lines 231-233). Authors should also include (in the supplementary material) an image of the HRM melting curves.

Thanks for the comment. We agree that information on the temperature ramp rate was missing. We have added the following sentence (L230): “The temperature ramp rate was 1.6°C/s.”

We have also added a picture of the HRM melting curves in the supplementary material, as suggested by Reviewer#3.

- Please carefully check spelling throughout the manuscript.

Ok, some typos and inconsistencies were present in the submitted manuscript. Following similar comments of Reviewer #1 and Reviewer #2, we have reviewed the entire manuscript.

---

## [Decision Letter · Decision Letter 1]

26 Feb 2024

Insect population dynamics under *Wolbachia*-induced cytoplasmic incompatibility: puzzle more than buzz in *Drosophila suzukii*

PONE-D-23-25628R1

Dear Dr. Auguste,

We’re pleased to inform you that your manuscript has been judged scientifically suitable for publication and will be formally accepted for publication once it meets all outstanding technical requirements.

Kind regards,

Bilal Rasool, PhD

Academic Editor

PLOS ONE

**Comments to the Author**

1. If the authors have adequately addressed your comments raised in a previous round of review and you feel that this manuscript is now acceptable for publication, you may indicate that here to bypass the “Comments to the Author” section, enter your conflict of interest statement in the “Confidential to Editor” section, and submit your "Accept" recommendation.

Reviewer #1: All comments have been addressed

Reviewer #2: All comments have been addressed

Reviewer #3: All comments have been addressed

2. Is the manuscript technically sound, and do the data support the conclusions?

Reviewer #1: Yes

Reviewer #2: Yes

Reviewer #3: Yes

3. Has the statistical analysis been performed appropriately and rigorously? 

Reviewer #1: Yes

Reviewer #2: Yes

Reviewer #3: Yes

4. Have the authors made all data underlying the findings in their manuscript fully available?

Reviewer #1: Yes

Reviewer #2: Yes

Reviewer #3: Yes

5. Is the manuscript presented in an intelligible fashion and written in standard English?

Reviewer #1: Yes

Reviewer #2: Yes

Reviewer #3: Yes

6. Review Comments to the Author

Reviewer #1: The manuscript has been substantially improved and it now reads much better than the previous version. The authors addressed all comments and recommendations. I fully recommend it for publication.

Reviewer #2: The manuscript is already improved after revision and may be considered for publication. However the journal's style formatting? typos and other inconsistencies, missing italics, missing information, use of abbreviations, double spaces etc may be rechecked before the final acceptance.

Reviewer #3: Authors have addressed my comments in a satisfactory way. Congratulations for the excellent study! I look forward to read also articles about the follow up experiments.

7. PLOS authors have the option to publish the peer review history of their article (what does this mean?). If published, this will include your full peer review and any attached files.

Reviewer #1: No

Reviewer #2: No

Reviewer #3: No

---

## [Editor Report · Acceptance letter]

1 Mar 2024

PONE-D-23-25628R1 

PLOS ONE

Dear Dr. Auguste, 

I'm pleased to inform you that your manuscript has been deemed suitable for publication in PLOS ONE. Congratulations! Your manuscript is now being handed over to our production team.

Kind regards, 

on behalf of

Dr Bilal Rasool 

Academic Editor

PLOS ONE